# Cryopreservation of Testicular Tissue from Adult Red-Rumped Agoutis (*Dasyprocta leporina* Linnaeus, 1758)

**DOI:** 10.3390/ani12060738

**Published:** 2022-03-16

**Authors:** Andréia M. Silva, Ana G. Pereira, Luana G. P. Bezerra, Samara S. Jerônimo Moreira, Alexsandra F. Pereira, Moacir F. Oliveira, Pierre Comizzoli, Alexandre R. Silva

**Affiliations:** 1Laboratory of Animal Germplasm Conservation, Department of Animal Sciences, Federal University of Semiarid Region–UFERSA, Mossoró 59625-900, RN, Brazil; andreia.m.silva@hotmail.com (A.M.S.); anagloriavet@gmail.com (A.G.P.); luana_grasielly@yahoo.com.br (L.G.P.B.); samara.sandy@bol.com.br (S.S.J.M.); alexsandra.pereira@ufersa.edu.br (A.F.P.); moacir@ufersa.edu.br (M.F.O.); 2Smithsonian Conservation Biology Institute, National Zoological Park, Veterinary Hospital, Washington, DC 20008, USA; comizzolip@si.edu

**Keywords:** biobanking, testicular tissue, wild rodents, vitrification, permeating cryoprotectants

## Abstract

**Simple Summary:**

Testicular tissues are composed of many types of germ cells, including early stages which can be grown up to fully formed spermatozoa following grafting or in vitro culture. The systematic banking of testicular tissues at freezing temperatures is useful for future use in assisted reproduction and to improve the reproductive management of rare mammalian species. The present study explored testicular tissue cryopreservation in the red-rumped agouti by slow freezing or vitrification methods, using different combinations of cryoprotectants. Solid-surface vitrification using the combination of dimethyl sulfoxide and ethylene glycol was the most effective protocol to preserve testicular cell morphology and proliferative potential.

**Abstract:**

This study measured the effects of different freezing techniques and permeating cryoprotectants on the preservation of testicular tissues from adult red-rumped agoutis. Tissue biopsies (3.0 mm^3^) from five individuals were allocated to different experimental groups: control (non-cryopreserved); slow freezing (SF), solid-surface vitrification (SSV), and conventional vitrification (CV). Each method used dimethyl sulfoxide (DMSO), ethylene glycol (EG), or a DMSO + EG combination. Morphology, viability, mitochondrial activity, and proliferative potential were assessed in fresh and frozen tissue samples. Testicular morphology was better using SSV with a combination of DMSO and EG. Across the different cryopreservation approaches, as well as cryoprotectant combinations, cell viability was comparable. Regarding mitochondrial activity, DMSO + EG/SSV or CV, and DMSO + EG/CV were similar to the EG/SF group, which was the best group that provided values similar to fresh control groups. Adequate preservation of the proliferative potential of spermatogonia, Leydig cells, and Sertoli cells was obtained using SSV with DMSO + EG. Overall, the use of SSV with DMSO + EG was the best protocol for the preservation of testicular tissues from adult red-rumped agoutis.

## 1. Introduction

The genus Dasyprocta, a wild hystricognath rodent, is composed of 13 different species of agoutis found throughout the Neotropical Americas [1]. These animals are important for the ecological balance, acting as seed dispersers and forming part of the food chain [2,3]. Among agoutis, the breeding of the red-rumped agouti (*Dasyprocta leporina*) is easy because it can adapt to captivity [4]. Thus, it is an excellent experimental model for the conservation of endangered hystricognath rodents, such as the *D. ruantanica* [5] and the *D. mexicana* [6].

Various efforts have been conducted to optimize assisted reproductive techniques and the preservation of agouti germplasms [7,8,9]. Thus far, the development of protocols for the cryopreservation and in vitro culture of ovarian tissues has been reported [8]. However, there are no data on agouti testicular tissues. Cryopreserving the male gonadal tissues enables the storage of germ cells at various developmental stages, including undifferentiated spermatogonia that can potentially provide an unlimited sperm production after grafting or in vitro culture of tissues [10,11]. Unlike laboratory rodents, the testes of agoutis are intra-abdominal, being subjected to peculiar physiological mechanisms that could lead to the need for adapted conservation protocols [12].

In mice and pigs, testicular tissue preservation is sufficiently effective to allow the production of viable offspring using sperm produced by the culture of cryopreserved tissues [11,13]. In wild species, recent publications have reported initial steps of the technology, especially in terms of the cryopreservation techniques and comparisons of different cryoprotectants [14,15]. Due to species-specific differences, the progress of technology suffers from the lack of protocols for certain species, because in most cases, the direct extrapolation of protocols from closely related species is not possible [16]. Overall, cryopreservation protocols have been based on slow freezing [15] and vitrification [14], which are mainly differentiated by the rate of cooling and concentration of cryoprotectants. In parallel, various cryoprotectants—isolated or in combination—have been tested for this purpose. For laboratory murine rodents such as rats, the isolated use of dimethyl sulfoxide (DMSO) is highlighted for immature individuals, whereas ethylene glycol (EG) is suggested for the cryopreservation of testes in mature individuals [17]. For some wild species, however, the combination of these cryoprotectants leads to the adequate recovery of viable testicular cells after cryopreservation [14,15,18].

The objective of this study was to measure the effect of different cryopreservation protocols (in association with different cryoprotectants) on the histomorphology, viability, proliferative potential, and mitochondrial activity of testicular cells from the red-rumped agouti.

## 2. Materials and Methods

### 2.1. Animal Ethics and Husbandry

All experiments were approved by the Animal Ethics Committee of the Federal Rural University of the Semi-Arid Region (UFERSA; Opinion no. 11/2019) and Chico Mendes Institute for Biodiversity Conservation (Opinion no. 66618-3). All individuals were raised at the Center of Multiplication of Wild Animals, UFERSA (Mossoró, RN, Brazil; 5°10′ S, 37°10′ W), registered at the Brazilian Institute of Environment and Renewable Natural Resources (IBAMA, no. 1478912).

### 2.2. Testicular Tissue Collection and Experimental Design

Gonads from five adult males (2.8 ± 0.54-year-old; weighing 2.38 ± 0.29 kg) were collected as part of an annual program aiming at controlling the experimental population. Testis pairs (one pair for each experimental replicate) were thoroughly washed in saline solution (0.9% NaCl) and transferred to the laboratory within 30 min at 22 °C.

Gonads were dissected to remove the surrounding tissues and extensively washed three times in saline solution. Dissected testes then were cut in multiple small pieces of 3 mm × 1 mm × 1 mm (3 mm^3^). For each individual, a total of 120 small fragments were collected and randomly allocated to 1 of the 10 following experimental groups: no cryopreservation (control); exposure to DMSO, EG, or DMSO + EG followed either by slow freezing (SF), solid-surface vitrification (SSV), or conventional vitrification (CV). In each treatment group (n = 12 testicular fragments per group), three fragments were evaluated for histomorphometry, cell viability, mitochondrial activity, and cell proliferative capacity.

### 2.3. Slow Freezing Solution and Method for Testicular Tissue Fragments

Minimum essential medium (MEM) supplemented with 0.25 M sucrose and 10% fetal bovine serum (FBS) served as the base of the SF solution. Cryoprotectants were added at the following concentrations: 1.5 M DMSO, 1.5 M EG, or 0.75 M DMSO + 0.75 M EG. In each treatment group, 12 tissue pieces were placed in 2.0 mL cryovial (Thermo Fisher Scientific, Pittsburgh, PA, USA) containing 2.0 mL of SF solution at 21 °C. Each cryovial was transferred to a Nalgene freezing container (Mr. Frosty^®^, (Thermo Scientific Nalgene, Rochester, NY, USA)) filled with isopropyl alcohol at 25 °C. The container was kept in a −80 °C ultralow freezer for 12 h. This system usually ensures a cooling rate of about −1 °C/min. After this first period, samples were plunged into a liquid nitrogen container for a one-week storage period [19].

### 2.4. Vitrification Solutions and Methods for Testicular Tissue Fragments

Minimum essential medium (MEM) supplemented with 0.25 M sucrose and 10% fetal bovine serum (FBS) served as the base of the vitrification solution. Cryoprotectants were added at the following concentrations: 3 M DMSO, 3 M EG, or 1.5 M DMSO + 1.5 M EG. Tissue fragments were exposed to vitrification solution for 5 min at 25 °C, following our previous protocols [14]. After removal of the excess solution on an aseptic absorbent filter, tissues were vitrified by conventional (CV) or solid-surface vitrification (SSV). For CV, tissue fragments were placed in cryotubes and plunged directly into liquid nitrogen for a one-week storage period [20]. Regarding SSV, fragments were deposited on an aluminum sheet for 30 s (solid surface of high thermal conductivity) in contact with liquid nitrogen. They were then transferred to cryotubes immersed in liquid nitrogen and stored for one week [14].

### 2.5. Warming of Cryopreserved Tissue Fragments

Cryovials were pulled out of the liquid nitrogen, kept in air for 1 min at 25 °C, and immersed for 2 min in a water bath at 37 °C. To remove the cryoprotectants, all testicular tissue fragments were thoroughly rinsed three times for 5 min in MEM with 10% FBS, and then exposed stepwise to decreasing concentrations of sucrose (0.50, 0.25 M, and 0 M sucrose) [14].

### 2.6. Histomorphology of Testicular Tissue Fragments

Tissue fragments from control and cryopreserved groups were fixed in Bouin solution for 12 h, embedded in paraffin blocks, sectioned (5.0 µm thickness), stained with hematoxylin–eosin, and evaluated under a light microscope (Olympus CX 31 RBSFA, Tokyo, Japan). Table 1 specifies the different parameters and scoring systems that were used to describe the histology. For each parameter, 5 seminiferous tubules in 6 different fields were evaluated, which corresponded to a total of 30 seminiferous tubules for each treatment [14]. Testicular tissue fragments with a total score of 3 were considered as morphologically normal. However, fragments with a total score of 0 were considered as degraded/degenerated.

### 2.7. Cell Viability in Testicular Tissue Fragments

Cells were dissociated and isolated from tissue fragments by enzymatic digestion [15]. Briefly, fragments were incubated with 0.2% collagenase type IV at 37 °C for 10 min, followed by the addition of an equal volume of FBS to stop the enzyme action. After centrifugation at 114× *g* for 5 min, the cell pellets were resuspended and incubated in a mixture of 3 µL propidium iodide (0.5 mg/mL in phosphate-buffered saline, PBS) and 5 µL Hoechst 33342 (40 µg/mL in PBS) for 10 min at 37 °C to detect viable testicular cells (red staining) and count the total of cells (blue nuclear staining), respectively. A total of 100 cells were counted and classified in each treatment group.

### 2.8. Cell Mitochondrial Activity in Testicular Tissue Fragments

The mitochondrial activity was determined according to the methodology adapted from Faure et al. [21]. A volume of 5 µL (500 µmol/L) of MitoTracker™ Red CMXRos was added to each cell suspension, briefly dissociated for the viability assay, and incubated for 15 min at 37 °C; then, cells were examined using standard immunofluorescence microscopy. The surface and intensity of mitochondrial staining was quantified through pixel count, using image analysis software (ImageJ, v 1.48, NIH, Bethesda, MD, USA). The intensity of fresh samples was used as a reference. Values obtained in each treatment group were divided by the intensity of fresh samples to obtain relative expression levels (arbitrary fluorescence units).

### 2.9. Cell Proliferative Potential in Testicular Tissue Fragments

Using a silver staining technique previously reported [14], the proliferative capacity potential was evaluated by detecting nucleolar organizer regions (NORs) in spermatogonia and Sertoli cells. Cells were identified through their morphological features (nuclear morphology and localization), as previously reported in agoutis [22]. Tissue sections mounted on slides were exposed to a silver solution composed of 1 part of 2% gelatin (in 1% aqueous formic acid) and 2 parts of 50% aqueous silver nitrate solution for 30 min in a dark room. After extensive washing in 5% thiosulfate solution for 10 min [23], the number of NOR dots was counted within the nucleoli of spermatogonia and Sertoli cells [22]. Observations were made in 10 randomly selected nuclei in 10 different fields at 1000× magnification, which resulted in 200 cells for each treatment group [24].

### 2.10. Statistical Analysis

Data are expressed as the means ± standard error of means (SEM). Values were first tested for normality (Shapiro–Wilk test) and homoscedasticity (Levene test). Non-parametric data were transformed into arcsine values before analysis. The effects of the cryopreservation technique and cryoprotectant combination on testicular parameters were analyzed by ANOVA. Tukey’s test was used for pair-wise comparisons of treatment groups. All analyses were performed using SAS, version 8.0 (SAS Institute, Inc., Cary, NC, USA).

For histomorphometry, scores were compared among all treatments with the Kruskal–Wallis test for independent samples using SPSS, version 22.0 (SPSS Inc., Chicago, IL, USA). For all analyses, differences were considered significant when *p*-values were < 0.05.

## 3. Results

### 3.1. Testicular Histomorphology

Most treatments prevented tubular cell swelling, tubular cell loss, rupture from basal membrane, and shrinkage from basal membrane parameters. Scores ranged from 2.79 to 2.96, which was similar to observations in control groups (*p* > 0.05) (Table 2, Figure 1). However, the use of CV with DMSO or EG led to scores that were lower than in control groups (*p* < 0.05) for tubular cell swelling (Figure 1I), tubular cell loss (Figure 1H), and rupture from basal membrane (Figure 1C,F,I; Table 2). Shrinkage from the basal membrane was most evident using SSV with DMSO (Figure 1E; Table 2) (*p* < 0.05). Regarding tubular structure, all treatments led to score values lower than control groups (*p* < 0.05) (Table 2).

### 3.2. Testicular Cell Viability

After all cryopreservation treatments, the percentages of cell viability were lower than in control groups (80.6 ± 0.5%; *p* < 0.05; Figure 2). There were no differences within cryopreservation treatments (*p* > 0.05), with percentages ranging from 31.6 ± 1.2% to 45.6 ± 2.9% (Figure 2).

### 3.3. Testicular Cells Mitochondrial Activity

Mitochondrial activity was similar in the EG/SSV group (0.90 ± 0.11%) and in the control group (1.00 ± 0.03%; *p* > 0.05; Figure 3). Overall, a reduction in mitochondrial activity was observed for all the other treatments in comparison with the control group (*p* < 0.05; Figure 3), with the lowest values observed in the DMSO/SSV group (0.29 ± 0.08%). Intermediate values of mitochondrial activity were observed with EG/SF, DMSO + EG/SSV, DMSO/CV, and DMSO + EG/CV (*p* > 0.05; Figure 3).

### 3.4. Testicular Cell Proliferative Potential

Average values of NORs (Figure 4A) were 2.92 ± 0.08 for spermatogonia (Figure 4B), 2.36 ± 0.07 for Leydig cells (Figure 4C), and 3.73 ± 0.11 NORs for Sertoli cells (Figure 4D). Preservations of spermatogonia proliferative potential in DMSO + EG/SF and DMSO + EG/SSV groups were comparable to control groups (*p* > 0.05; Figure 4B), whereas values increased (*p* < 0.05) in all the other treatment groups. Preservations of Leydig cell proliferative potential in DMSO/SF and DMSO + EG/SSV groups were similar to control groups (*p* > 0.05; Figure 4C). Reductions (*p* < 0.05) in the NOR values were observed for the other groups, except for DMSO/SSV group, which exhibited a significant increase (*p* < 0.05) in the number of NORs in comparison to the control group (Figure 4C). Preservations of Sertoli cell proliferative potentials in DMSO + EG/SF, DMSO + EG/SSV and EG/CV groups were similar to the control groups (*p* > 0.05; Figure 4D), whereas value increases in the proliferative potential were found in all the other treatment groups (*p* < 0.05; Figure 4D).

## 4. Discussion

The collective results provide the first steps on the development of technology for hystricognath rodents, using adult red-rumped agoutis as scientific models, and highlighting the efficiency of SSV with DMSO + EG. This is different from what was postulated for rats, a murine rodent for which the isolated use of EG associated with slow freezing is suggested for adult individuals, and DMSO for immature individuals [17]. Although both species belong to the Rodentia order, there are marked anatomical and physiological differences in gonads between adult rats and agoutis. Rat testicles are oval-shaped and located in the scrotum [25], whereas the gonads of male agouti are craniocaudally elongated and located in an intra-abdominal position [12]. Testicular cells from both species are subjected to completely different physiological environments and temperatures, which could be the reason why distinct cryopreservation protocols are needed.

Regardless of the freezing technique used, the DMSO + EG combination efficiently preserved four out of five histomorphological parameters evaluated, preventing tubular cell swelling and loss, as well as rupture and shrinkage from the basal membrane. Regarding the overall testicular tubular structure, the DMSO + EG combination as well as the isolated cryoprotectants failed to provide efficient protection. Adult testicular tissues are sensitive to manipulations because they contain different cell types in seminiferous tubules, each with complex metabolisms [26]. Thus, the use of cryoprotectant combinations is recommended, mainly to diminish the toxic effects of single cryoprotectants on tissue structures, especially with the use of vitrification techniques which require high concentrations of cryoprotectants [27]. This combination of cryoprotectants was previously proven to be efficient for the preservation of testicular tissue histomorphology of adult collared peccaries [14]. The two abovementioned cryoprotectants seem to act synergistically. The exposure to DMSO leads to cell dehydration and prevents the intracellular formation of ice crystals during cryopreservation. It also interacts with the lipid membrane and induces pore formation for the passage of water [28]. Additionally, EG reduces the formation of ice crystals due to its capacity to penetrate the cell and replace intracellular water, balancing intracellular proteins and preventing membrane rupture [23,29].

Using DMSO alone, but only when SF was conducted, four out of five histomorphological parameters were well conserved after agouti testicular tissue preservation, except for the tubular cell structure. With a lower concentration than for vitrification, DMSO alone was able to provide a gradual exchange of water under a slow temperature decrease enabled by SF [19]. On the other hand, the use of EG, especially during CV, severely impaired most of the histomorphological parameters of agouti testicular tissue. Probably, the high EG concentration damaged the testicular cells, because it is known that this cryoprotectant generates potentially toxic byproducts, such as glycolic acid and oxalate, which promote an accumulation of these compounds in the intracellular environment [30].

Regarding cell viability, fresh agouti testicular tissues presented around 80.6% viable cells. After cryopreservation, there was a decrease in cell viability, because values from 45.6% to 31.6% viable cells were found; however, there were no differences among distinct treatments. The present results are similar to those found in the literature; the majority of protocols currently used for testicular tissue preservation are reported to provide low to moderate proportions of cell viability in other adult rodents, such as Holtzman rats (Rattus norvegicus: ~40%) [17]. In contrast, testicular tissues from immature individuals seem to be more resistant to freezing processes, thus presenting higher proportions of viable cells, as observed for mice (Mus musculus: ~90%) [31].

Overall, the mitochondrial activity in agouti testicular tissues was better preserved with vitrification techniques than SF. Compared with SF, vitrification is considered more efficient to prevent the intracellular and extracellular formation of ice crystals. There is a lower risk of mechanical damage to cells and organelles such as mitochondria because of the formation of a solidified amorphous state due to high-speed cooling and high concentrations of permeating cryoprotectants [32,33]. Moreover, temperature variations during cryopreservation procedures could contribute to mitochondrial disfunctions, especially with SF protocols using a Nalgene container [19].

As observed with the use of the DMSO + EG combination in both vitrification techniques, the use of EG alone during SSV exerted adequate preservation of mitochondrial activity in agouti testicular tissues. The low toxicity of EG, associated with its high permeability [34] and its low molecular weight (62.07 g/mol), allows its rapid influx during equilibration and dilution [35]. Probably, the rapid influx of EG in testicular cells contributed to the preservation of mitochondrial activity, even when it was combined with DMSO, especially acting on the preservation of glutamate + malate-supported respiration, as also described during preservation of muscle fibers [36]. On the other hand, the isolated use of DMSO in both SF and SSV yielded the lowest values for mitochondrial activity. This could be a consequence of the pro-oxidant ability of DMSO [30], whose oxidative stress can cause lipid peroxidation [37], which alters the fluidity of membranes, reducing the selectivity in ion transport and transmembrane signaling, which impairs cell transport [38].

Regarding the cell proliferative potential, the use of a DMSO + EG combination during SSV was the only treatment preserving the number of NORs in spermatogonia, Leydig and Sertoli cells, with values comparable with those observed for the fresh control. Due to the close relationship between nucleolar organizer regions (NORs) and cellular activity, NOR size and number can reflect or predict cell proliferation, transformation, or evident malignancy [39]. After cryopreservation, it is expected that NORs remain similar to the controls, only rising after an in vitro culture as a response of tissue developments. This would occur as a response to the condensation of chromosomes in mitosis, because some of the proteins involved in the transcription of the rDNA sites remain attached to these sites, resulting in the under-condensation of these chromosomal regions and the formation of secondary constrictions, also called nucleolar organizer regions (NORs) [40]. In the other treatments, abnormal cell activity would be probably related to an increase in NORs, thus reflecting a malignant activation state [39].

## 5. Conclusions

In conclusion, the collective results indicate the use of a cryoprotectant combination composed of DMSO and EG associated with a solid-surface vitrification technique for the cryopreservation of testicular tissues derived from adult red-rumped agoutis. To the best of our knowledge, this is the first description of an effective preservation of male gonadal tissues in this species, representing an initial step for the adaptation of this methodology to wild hystricognath rodent preservation.

## Figures and Tables

**Figure 1 animals-12-00738-f001:**
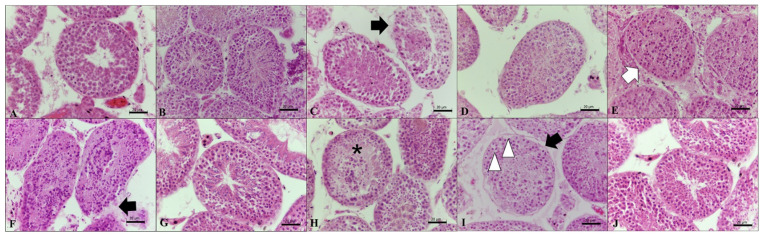
Representative micrographs used for histomorphological evaluations of testicular tissue sections from adult red-rumped agoutis (n = 5 males) after different cryopreservation treatments. (**A**) Non-cryopreserved control group, and cryopreserved groups using (**B**) slow freezing [SF] with dimethyl sulfoxide [DMSO], (**C**) SF with ethylene glycol [EG], (**D**) SF with DMSO + EG, (**E**) solid-surface vitrification [SSV] with DMSO, (**F**) SSV with EG, (**G**) SSV with EG + DMSO, (**H**) conventional vitrification [CV] with DMSO, (**I**) CV with EG, (**J**) CV with DMSO + EG. Black arrows indicate rupture from the basal membrane. White arrow shows shrinkage from the basal membrane. Asterisks shows tubular cell loss. White arrowheads show cell swelling. Scale bar: 20 µm.

**Figure 2 animals-12-00738-f002:**
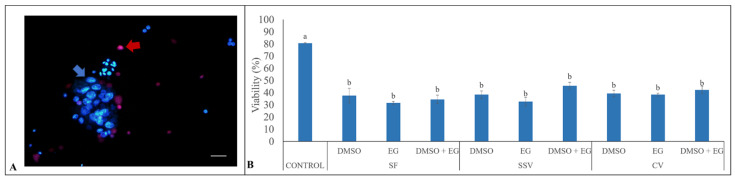
Evaluations of cell viability in testicular tissues from adult red-rumped agoutis (n = 5 males) after different cryopreservation treatments. (**A**) Representative picture of testicular cell viability evaluated by fluorescent probes. Blue arrow indicates cells in suspension stained with Hoechst 33342; red arrow indicates non-viable cells in the same suspension stained with propidium iodide. (**B**) Average proportions (mean ± SEM) of viable cells in testicular tissues exposed to different treatments (slow freezing, SF; conventional vitrification, CV; and solid-surface vitrification, SSV) and different cryoprotectants: dimethyl sulfoxide (DMSO), ethylene glycol (EG), and DMSO + EG. Scale bar: 20 µm. ^a,b^ Different lowercase letters above bars indicate differences among treatments (*p* < 0.05).

**Figure 3 animals-12-00738-f003:**
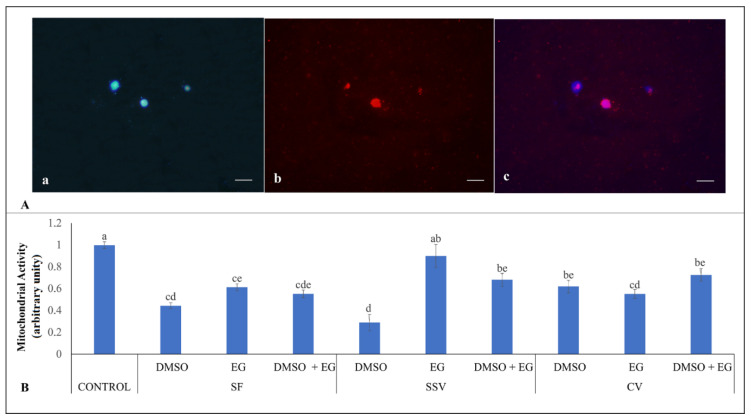
Evaluations of mitochondrial activity (MitoTracher Red staining) in testicular tissues from adult red-rumped agoutis (n = 5) after different cryopreservation treatments. (**A**) Representative images for: (**a**) Hoechst 33342 staining, (**b**) MitoTracher Red staining, (**c**) merged images; (**B**) Average values (mean ± SEM) of arbitrary unities used for the evaluation of mitochondrial activity in testicular tissues exposed to different treatments (slow freezing, SF; conventional vitrification, CV; and solid-surface vitrification, SSV) and different cryoprotectants: dimethyl sulfoxide (DMSO), ethylene glycol (EG), and DMSO + EG. Scale bar: 20 µm. ^a–e^ Different lowercase letters above bars indicate differences among treatments (*p* < 0.05).

**Figure 4 animals-12-00738-f004:**
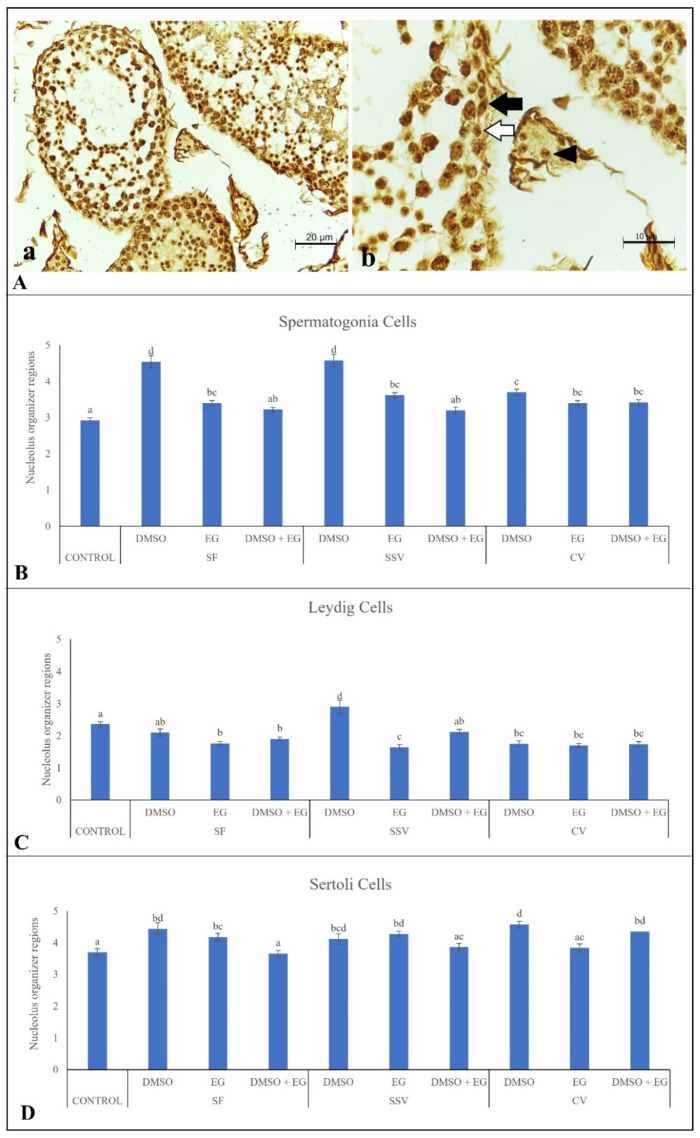
(**A**) Representative picture of proliferative potential evaluated by the quantification of nucleolar organizer regions (NORs) in testicular cells from adult red-rumped agoutis (n = 5 males): (**a**) Seminiferous tubule, scale bar: 20 µm; (**b**) enlarged seminiferous tubule area showing spermatogonia (white arrow), Leydig cell (black arrowhead), and Sertoli cell (black arrow), scale bar: 10 µm. (**B**–**D**) evaluations of proliferative capacity as the average number NORs (mean ± SEM) in spermatogonia (**B**), Leydig cells (**C**), and Sertoli cells (**D**) exposed to different treatments (slow freezing, SF; solid-surface vitrification, SSV; and conventional vitrification, CV) with different cryoprotectants (dimethyl sulfoxide, DMSO; ethylene glycol, EG; DMSO + EG combination). ^a–d^ Different lowercase letters above bars indicate differences among treatments (*p* < 0.05).

**Table 1 animals-12-00738-t001:** Morphological parameters of seminiferous tubules from the testes of red-rumped agoutis.

Parameter	Scores		
	#3	#2	#1
Tubular cell swelling	No swelling	>50% cells without swelling	>50% cells with swelling
Tubular cell loss	No cell loss	<75% cell types lost	>75% cell types lost
Rupture from basal membrane	No rupture	Partly ruptured (<50%)	Mostly ruptured (>50%)
Shrinkage from basal membrane	No shrinkage	Partly shrinkage (<50%)	Mostly shrinkage (>50%)
Tubular structure	Intact structure	All cell types present although slightly disordered structure	Random distribution of remaining cells

**Table 2 animals-12-00738-t002:** Morphological evaluation (3, normal to 1, degenerated) of adult *Dasyprocta leporina* (n = 5 males) fresh (control) vs. cryopreserved testicular tissues using different cryopreservation techniques (slow freezing, SF; conventional vitrification, CV; solid surface vitrification, SSV) and cryoprotectants (dimethyl sulfoxide (DMSO); ethylene glycol (EG)).

		Tubular Cell Swelling	Tubular Cell Loss	Rupture from Basal Membrane	Shrinkage from Basal Membrane	Tubular Structure
Control		2.80 ± 0.04 ^a,b^	2.91 ± 0.03 ^a,b^	2.91 ± 0.02 ^a,b^	2.94 ± 0.02 ^a,b^	2.36 ± 0.04 ^a^
SF	DMSO	2.88 ± 0.03 ^a,b^	2.93 ± 0.02 ^a,b^	2.92 ± 0.03 ^a,b,c^	2.88 ± 0.03 ^a,b^	2.04 ± 0.02 ^b^
	EG	2.80 ± 0.04 ^a,b^	2.86 ± 0.03 ^a,b^	2.88 ± 0.03 ^c^	2.76 ± 0.04 ^a,c^	2.03 ± 0.02 ^b^
	DMSO + EG	2.83 ± 0.04 ^a,b^	2,92 ± 0.03 ^a,b^	2.95 ± 0.02 ^a,b^	2.96 ± 0.02 ^a^	2.02 ± 0.01 ^b^
SSV	DMSO	2.88 ± 0.03 ^a^	2.85 ± 0.03 ^b,c^	2.77 ± 0.03 ^b,c^	2.87 ± 0.03 ^c^	2.02 ± 0.01 ^b^
	EG	2.85 ± 0.03 ^a,b^	2.92 ± 0.02 ^a,b^	2.87 ± 0.03 ^b,c^	2.88 ± 0.03 ^a,c^	2.01 ± 0.01 ^b^
	DMSO + EG	2.94 ± 0.02 ^a^	2.99 ± 0.01 ^a^	2.95 ± 0.02 ^a^	2.98 ± 0.01 ^a^	2.03 ± 0.02 ^b^
CV	DMSO	2.73 ± 0.04 ^b,c^	2.93 ± 0.02 ^a,b^	2.91 ± 0.03 ^b,c^	2.84 ± 0.03 ^a,b^	2.04 ± 0.02 ^b^
	EG	2.65 ± 0.04 ^c^	2.78 ± 0.04 ^c^	2.79 ± 0.04 ^c^	2.76 ± 0.04 ^b,c^	2.09 ± 0.03 ^b^
	DMSO + EG	2.96 ± 0.02 ^a^	2.94 ± 0.02 ^a,b^	2.88 ± 0.04 ^a,b^	2.98 ± 0.02 ^a,c^	2.02 ± 0.02 ^b^

^a–c^ Different superscript lowercase letters indicate significant differences among treatments according to Kruskal–Wallis tests (*p* < 0.05).

## Data Availability

The data presented in this study are available on request from the corresponding author.

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
