# Peer review of "Cryopreservation of Testicular Tissue from Adult Red-Rumped Agoutis (Dasyprocta leporina Linnaeus, 1758)"

_animals, 2022, doi:10.3390/ani12060738_

Round 1

Reviewer 1 Report

  1. The title of the article should be simplified to make it clear.
  2. In statistical analysis, the results are presented in a way that should be indicated in the article, Mean ± SEM or mean ± SD?
  3. Line 123 How much time in the water bath.
  4. Line 160 It is well known that mature Sertoli cells in the testis have no proliferative capacity, and the authors' assay of the proliferative capacity of Sertoli cells does not seem to have any basis.
  5. Line 227-228 EG/SSV group (0.90 ± 0.11), control group (1.00 ± 0.03%), the format of the data presented in paper should be uniform.
  6. Line 245 In testicular cell proliferative potential, there are many cell types in testis, the authors only determine the cell type by the location of the cells is not a rigorous method, the authors should consider a convincing method to show that the tested cells are the target cells.
  7. Line 311-310, this part of the discussion is insufficient, and the part of the discussion should be rewrote.

Author Response

  1. The title of the article should be simplified to make it clear.

Answer: We changed the title to “Cryopreservation of testicular tissue from adult red-rumped agouti (Dasyprocta leporina Linnaeus, 1758)”.

  1. In statistical analysis, the results are presented in a way that should be indicated in the article, Mean ± SEM or mean ± SD?

Answer: We added that “Data were expressed as means ± standard error of means”.

  1. Line 123 How much time in the water bath.

Answer: We specified that “the cryovials were immersed for 2 min in a water bath at 37 °C”.

  1. Line 160 It is well known that mature Sertoli cells in the testis have no proliferative capacity, and the authors' assay of the proliferative capacity of Sertoli cells does not seem to have any basis.

Answer: Thank you for raising that point. Indeed, mature Sertoli cells in the testicles usually do not proliferate. However, it has been shown in in vitro studies that adult Sertoli cells from mice and men [Ahmed et al., Biol Reprod 2009; 80:1084–1091] and in hamsters [Tarulli et al., Biol Reprod 2006; 74:798–806.] that Sertoli cells can regain their proliferative ability. The explanation is that molecular mechanisms needed to reattain proliferative potential of Sertoli cells in higher mammals may have persisted and could be elicited with appropriate stimuli [Tarulli et al., Biol Reprod 2012; 87(1):13, 1–11]. We used the AgNor technique to identify the nucleolar organizer regions (NORs), which size and numbers can reflect or predict cell proliferation, transformation, or evident malignancy potential. Therefore, this analysis was conducted to identify if cryopreservation process would induce abnormal number of NORs that would be indicative of abnormalities. Our results then demonstrated that preservation of Sertoli cell proliferative potential in DMSO + EG/SF, DMSO + EG/SSV and EG/CV groups were like in control groups (P> 0.05; Figure 4D), while values increased were found in all the other treatment groups (indicating possible abnormalities).

  1. Line 227-228 EG/SSV group (0.90 ± 0.11), control group (1.00 ± 0.03%), the format of the data presented in paper should be uniform.

Answer: We apologize for the mistake, we corrected that.

  1. Line 245 In testicular cell proliferative potential, there are many cell types in testis, the authors only determine the cell type by the location of the cells is not a rigorous method, the authors should consider a convincing method to show that the tested cells are the target cells.

Answer: We considered the cell characteristics and localization into the seminiferous tubule according to the classification of the different cell types reported by Costa et al. (J. Androl. 2010, 31, 489–499, oi:10.2164/JANDROL.109.009787) for red-rumped agoutis.

  1. Line 311-310, this part of the discussion is insufficient, and the part of the discussion should be rewrote.

Answer: We rewrote the sentence as requested “After cryopreservation, there was a decrease on cell viability, since values from 45.6% to 31.6% viable cells were found; however, there were no differences among distinct treatments.”

Reviewer 2 Report

Cryopreservation of the gonads is important for human and animals, especially for the biodiversity conservation in wild animals. This manuscript investigated this issue by using adult red-rumped agoutis as the model. The scientific design, methodology, data, and the interpretation of data are all good. And the article was written in good English. 

Line 4 in 2.3, "12 tissue pieces we placed in" should be "12 tissue pieces were placed in".

Author Response

Reviewer #2

  1. Cryopreservation of the gonads is important for human and animals, especially for the biodiversity conservation in wild animals. This manuscript investigated this issue by using adult red-rumped agoutis as the model. The scientific design, methodology, data, and the interpretation of data are all good. And the article was written in good English.

Answer: Authors thank the reviewer for the comments.

  1. Line 4 in 2.3, "12 tissue pieces we placed in" should be "12 tissue pieces were placed in".

Answer: We correct it as requested.

Reviewer 3 Report

This is a well-conducted research focused in the development of an efficient cryopreservation procedure to preserve testis samples from adult red-rumped agoutis. Despite the study being of high applied interest, it need some revision before its acceptance for publication, especially of the Figures.

  • Images provided in Figure 1 are too small, and so it is difficult to distinguish the different structures and cell types, and even the alteration induced by some protocols. Therefore, authors must provide a new Figure with high quality images that help the reader to understand the effect of the different treatments.
  • Authors must also improve the quality of Figure 2 by providing a high quality microscopic image and a high resolution plot. Please, note that the content of the plot is difficult to read. Moreover, regarding to legend of the plot, authors must specify the sample size.
  • Authors must also provide high resolution images for Figure 3A.
  • Authors must provide a title for Table 2. Please, note that the information given in the current title must be placed in the Table legend. Authors must also indicate the sample size.

Author Response

Reviewer #3

  1. This is a well-conducted research focused in the development of an efficient cryopreservation procedure to preserve testis samples from adult red-rumped agoutis. Despite the study being of high applied interest, it need some revision before its acceptance for publication, especially of the Figures.

Answer: Thank you for the comments. Regarding figures, the guide for authors ask us to include the images and tables into the journal template. When we did this at the previous version, all the images lost resolution and definition. For this reason, we decided to include figures and tables at the end of the manuscript in this new version for a better visualization.

  1. Images provided in Figure 1 are too small, and so it is difficult to distinguish the different structures and cell types, and even the alteration induced by some protocols. Therefore, authors must provide a new Figure with high quality images that help the reader to understand the effect of the different treatments.
  2. Authors must also improve the quality of Figure 2 by providing a high quality microscopic image and a high resolution plot. Please, note that the content of the plot is difficult to read. Moreover, regarding to legend of the plot, authors must specify the sample size.
  3. Authors must also provide high resolution images for Figure 3A.
  4. Authors must provide a title for Table 2. Please, note that the information given in the current title must be placed in the Table legend. Authors must also indicate the sample size.

Answers: All information requested related to figures or tables follows are in separate files at the end of the manuscript.
